# Surfactin and Capric Acid Affect the Posaconazole Susceptibility of *Candida albicans* Strains with Altered Sterols and Sphingolipids Biosynthesis

**DOI:** 10.3390/ijms242417499

**Published:** 2023-12-15

**Authors:** Daria Derkacz, Monika Grzybowska, Liliana Cebula, Anna Krasowska

**Affiliations:** Faculty of Biotechnology, University of Wroclaw, 50-383 Wroclaw, Poland; daria.derkacz@uwr.edu.pl (D.D.); monika.grzybowska00@gmail.com (M.G.); cebula.liliana@op.pl (L.C.)

**Keywords:** *Candida albicans*, posaconazole, surfactin, capric acid, azole resistance, synergism

## Abstract

Infections caused by *Candida* spp. pose a continuing challenge for modern medicine, due to widespread resistance to commonly used antifungal agents (e.g., azoles). Thus, there is considerable interest in discovering new, natural compounds that can be used in combination therapy with conventional antibiotics. Here, we investigate whether the natural compounds surfactin and capric acid, in combination with posaconazole, enhance the growth inhibition of *C. albicans* strains with alterations in sterols and the sphingolipids biosynthesis pathway. We demonstrate that combinations of posaconazole with surfactin or capric acid correspond with the decreased growth of *C. albicans* strains. Moreover, surfactin and capric acid can independently contribute to the reduced adhesion of *C. albicans* strains with altered ergosterol biosynthesis to abiotic surfaces (up to 90% reduction in adhesion). A microscopic study of the *C. albicans* plasma membrane revealed that combinations of those compounds do not correspond with the increased permeabilization of the plasma membrane when compared to cells treated with posaconazole alone. This suggests that the fungistatic effect of posaconazole in combination with surfactin or capric acid is related to the reduction in adhesion of *C. albicans*.

## 1. Introduction

Posaconazole (PSZ) is a third-generation triazole antifungal agent, which was approved by the Food and Drug Administration in 2006. It has been used in severely immunocompromised patients, and for the treatment of oropharyngeal candidiasis [1]. As has been observed for other azole derivatives, the mode of action of PSZ is based on the inhibition of activity of lanosterol 14α-demethylase (Erg11p, CYP51), leading to the accumulation of toxic methylated sterol precursors, and, thereby, the depletion of ergosterol from the plasma membrane (PM) [2]. This is associated with alterations in the fungal PM structure and the inhibited growth of the pathogen.

The chemical structure of PSZ consists of a triazole ring, a furan ring and two fluorine atoms (Figure 1). PSZ is characterized by the presence of an extended side chain that provides additional contact points with Erg11p [3]. Multiple hydrophobic contacts of the side-chain of PSZ may result in an increased binding affinity to the enzyme, thereby inhibiting its activity more effectively than first-generation azole drugs. It has been established that the 1,2,4-triazole ring, together with the 2,4-difluorophenyl substituent, are necessary to obtain the greatest antifungal activity, and the strongest potential of the drug in inhibiting Erg11p [4].

In contrast to most azole drugs, PSZ appears to have fungistatic or fungicidal activity, depending on the species [5,6]. For example, it has been reported that PSZ can exhibit activity against fluconazole-resistant *Candida* spp. (FLC) [7,8].

The common usage of azoles has contributed to azole-resistance among *Candida* spp., which complicates the treatment of patients suffering from candidiasis [9,10]. Therefore, searching for natural compounds that are active against fungal pathogens or that improve the activity of commonly used azoles remains a key research aim of modern medicine.

We previously reported that the simultaneous treatment of *C. albicans* with both FLC and surfactin (SU) in the mixture corresponds with decreased MIC50 values for FLC, along with increased PM permeabilization and the remodeling of the pathogen cell wall [11]. SU is a natural lipopeptide with an amphiphilic structure, produced by *Bacillus subtilis*. It possesses broad-spectrum antifungal and antibacterial activity [12,13]. Similar to other known natural surfactants (e.g., pseudofactin II), SU has an ability to reduce the adhesion of pathogens to surfaces, which results in a reduced ability to form biofilms, and decreases the hydrophobicity of the fungi [14].

Another promising compound that could be used as an additive in azoles treatments is capric acid (CA), which is naturally produced by the probiotic yeasts *Saccharomyces boulardii* [15]. It has been proven that CA independently impacts the growth, filamentation and biofilm formation of *C. albicans* [16]. Additionally, CA increases the susceptibility of *C. albicans* to both FLC and amphotericin B [17]. The chemical structures of SU and CA are presented in Figure 2.

Based on prior data, we aimed to determine if SU and CA will affect the susceptibility of *C. albicans* to another azole drug, PSZ. Since azoles influence the structure and functioning of the fungal PM, we considered how a combination of PSZ and SU or CA would impact the growth of *C. albicans* mutants with alterations in ergosterol (*C. albicans* KS058 strain, lacking ergosterol in the PM due to the full deletion of *ERG11,* and three *C. albicans* strains, 10C1B1I1, 27A5A33A and 9B4B34A, with amino-acid substitutions in Erg11p) or mutations impacting the sphingolipids (SLs) biosynthesis pathway (*C. albicans* strains lacking *FEN1*, *FEN12* or both genes). We also investigated whether perturbations in SLs content in the PM correspond with the altered susceptibility of these strains. This research is the continuation of our previous investigation into the synergistic activity of fungistatic azoles with SU or CA [11,17]. Considering that our earlier studies have already proven that alterations in ergosterol levels affect the susceptibility of *C. albicans* to those compounds, we wanted to determine whether the level of SLs could also impact the response to synergistic therapy. This research is an attempt at an explanation of the mechanism of action of the combination of PSZ with SU and CA. Such research can potentially impact the perception of combined therapies used in the treatment of *Candida*-related infections.

## 2. Results

### 2.1. Susceptibility of Candida albicans Strains with Alterations in Ergosterol and Sphingolipids Biosynthesis to Posaconazol in Combination with Surfactin or Capric Acid

#### 2.1.1. Susceptibility of *Candida albicans* Strains to Posaconazole in Combination with Surfactin or Capric Acid

We aimed to determine if mutations impacting sterol or SLs biosynthesis in *C. albicans* affect the susceptibility of the pathogen to PSZ, either alone or in combination with SU or CA.

##### Posaconazole in Combination with Surfactin

Figure 3 represents the growth of *C. albicans* strains with the deletion of the *ERG11* gene, or with amino acid substitutions in Erg11p, in the presence of PSZ, with or without SU.

In the case of *C. albicans* SC5314 (WT), 10C1B1I1, and 9B4B34A, the MIC50 value for PSZ was 0.0156 μg/mL (Appendix A), although the 10C1B1I1 strain was more susceptible to PSZ at a concentration of 0.0156 μg/mL than the WT strain (Figure 3A,C). The MIC50 for PSZ alone could not be determined for the *C. albicans* 27A5A33A strain over the tested range of PSZ concentrations (Appendix A), but a drop in growth (%) was observed at a higher concentration of PSZ (0.0313–0.125 μg/mL) when compared to the control condition (0 μg/mL PSZ).

The lack of production of Erg11p (a target for azoles) by the *C. albicans* KS058 strain resulted in the resistance of that strain to PSZ over all tested concentrations. The addition of SU at 4 and 8 μg/mL resulted in the increased growth of the strains. The presence of 32 μg/mL of SU resulted in the significant growth inhibition of *C. albicans* KS058 (Figure 3B).

One interesting result is that in the presence of PSZ (0.0156 μg/mL) with SU (4 μg/mL), we observed the increased growth of *C. albicans* WT, 10C1B1I1, 27A5A33A and 9B4B34A, compared to the growth of those strains in the presence of PSZ (0.0156 μg/mL) alone. Increasing the PSZ and SU concentrations resulted in a successive decrease in the growth of all tested strains.

Figure 4 represents the growth of *C. albicans* strains with deletions of genes involved in SLs biosynthesis in the presence of PSZ, with or without SU.

For *C. albicans* SN95 (WT), *fen1Δ/Δ* and *fen12Δ/Δ*, the MIC50 for PSZ was 0.0156 μg/mL, while for the *fen1Δ/Δ*;*fen12Δ/Δ* strain, the MIC50 was 0.0078 μg/mL (Appendix A). According to the data presented in Figure 2, *C. albicans fen1Δ/Δ* and *fen12Δ/Δ* were more resistant to a higher concentration of PSZ alone (0.0078–0.125 μg/mL) or in combination with SU (4–32 μg/mL) than the WT strain. The double deletant at the *FEN1* and *FEN12* genes was less susceptible to treatment with a combination of PSZ and SU than the *C. albicans* WT strain.

##### Posaconazole in Combination with Capric Acid

Figure 5 represents the growth of *C. albicans* strains with the deletion of the *ERG11* gene or with amino-acid substitutions in Erg11p, in the presence of PSZ, with or without SU.

Compared to *C. albicans* WT, the 10C1B1I1, 27A5A33A, and 9B4B34A strains were more susceptible to PSZ (0.0078 μg/mL) in combination with CA at both concentrations: 11.35 and 22.7 μg/mL. In the presence of PSZ and CA (0.0078 and 45.3 μg/mL, respectively), *C. albicans* 10C1B1I1 and 27A5A33A, but not 9B4B34A, were more resistant than the WT strain.

The growth of *C. albicans* KS058 in the presence of PSZ and CA was diminished by about 50% in comparison with growth under the control conditions (0 μg/mL of CA). Only the highest concentration of CA (90.6 μg/mL) resulted in almost complete growth inhibition of the strain.

Figure 6 depicts the growth of *C. albicans* strains with deletions of the genes involved in SLs biosynthesis, in the presence of PSZ, with or without CA.

*C. albicans fen1Δ/Δ* and *fen12Δ/Δ* strains were more resistant than the WT strain to combinations of PSZ and CA (0.0156 and 11.35, 22.7 μg/mL, respectively), but the increasing addition of CA reduced the difference in growth among *C. albicans* strains. The double deletion of the *FEN1* and *FEN12* genes resulted in the significant growth inhibition of *C. albicans* in the presence of 11.35 and 22.7 μg/mL of CA, while 45.30 and 90.60 μg/mL concentrations of CA led to complete growth inhibition of the *C. albicans fen1Δ/Δ*;*fen12Δ/Δ* strain. This suggests that for this strain, CA is more toxic than SU (Figure 4D).

#### 2.1.2. Growth of *Candida albicans* Strains in the Presence of Posaconazole in Combination with Surfactin or Capric Acid

The growth curves of *C. albicans* strains with the deletion of the *ERG11* gene or with amino acid *substitutions* in Erg11p, under control conditions (YPD) in the presence of PSZ (0.0156 μg/mL) and PSZ with either SU (4 μg/mL) or CA (45.3 μg/mL), are presented in Figure 7.

The addition of PSZ to *C. albicans* KS058 and 27A5A33A did not affect the growth of those strains. In the case of the *C. albicans* 10C1B1I1 strain, we observed diminished growth compared to control conditions. A shift in the logarithmic phase of growth of *C. albicans* 9B4B34A was observed in the presence of PSZ, and a similar trend in the growth curve was observed for the culture with the addition of PSZ and CA. Additionally, we noted that the presence of SU in the culture resulted in barely any growth of all the investigated strains. For *C. albicans* KS058, which lacks Erg11p, a target for azoles, we detected a significant decrease in growth in the presence of SU.

The growth curves of *C. albicans* strains with deleted genes involved in SLs biosynthesis, under control conditions (YPD) and in the presence of PSZ (0.0156 or 0.0078 μg/mL for *fen1Δ/Δ;fen12Δ/Δ* strain) and PSZ with SU (4 μg/mL) or CA (22.7 or 11.35 μg/mL for *fen1Δ/Δ;fen12Δ/Δ* strain), are presented in Figure 8.

The presence of PSZ in cultures of *C. albicans fen1Δ/Δ*, *fen12Δ/Δ* and *fen1Δ/Δ*;*fen12Δ/Δ* did not result in the decreased growth of those strains, in contrast to the WT strain. The SN95 (WT) strain, in the presence of PSZ, entered the logarithmic phase after 9–10 h of culture, while under control conditions, logarithmic growth started in the 7th hour of culture. In the presence of PSZ and CA, the growth curves of the *C. albicans fen1Δ/Δ* and *fen12Δ/Δ* strains remained similar to those obtained under control conditions, except for a shift in the timing of the logarithmic phase of growth by 2–3 h. The growth of SN95 (WT) and *fen1Δ/Δ*;*fen12Δ/Δ* was significantly reduced in the presence of PSZ and CA.

The combination of PSZ and SU resulted in barely any growth of the *C. albicans* SN95 (WT) strain. An interesting observation is that, for *fen1Δ/Δ*, *fen12Δ/Δ* and *fen1Δ/Δ*;*fen12Δ/Δ*, the growths of these strains were significantly reduced compared to those under control conditions, and after a short stationary phase of growth, we observed a decrease in OD_600_ value, which suggests that cells entered the death phase.

### 2.2. Candida albicans Strains with Altered Ergosterol or Sphingolipids Biosynthesis Exhibit Diminished Adhesion to Abiotic Surfaces in the Presence of Surfactin or Capric Acid

The adhesion to abiotic surfaces of all investigated *C. albicans* strains under control conditions, in the presence of SU (4 μg/mL) or CA (45.3; 22.7 or 11.35 μg/mL depending on *C. albicans* strain), is presented in Figure 9.

The presence of SU or CA resulted in the significantly decreased adhesion to abiotic surfaces of all *C. albicans* strains with the deletion of the *ERG11* gene, or with amino acid substitutions in Erg11p (Figure 9A), compared to results under control conditions (PBS alone). However, all of those strains were characterized by better adhesion to abiotic surfaces in the presence of CA, in comparison to SU. The highest reduction in adhesion was observed for *C. albicans* KS058 in the presence of SU, but it is worth mentioning that KS058 also exhibited reduced adhesion under control conditions compared to other strains, and the addition of SU enhanced that phenotype.

The *C. albicans* strains with deletions of genes involved in SLs biosynthesis exhibited reduced adhesion in the presence of SU, compared to under control conditions (Figure 9B). The *C. albicans fen1Δ/Δ*;*fen12Δ* strain, acting similarly to *C. albicans* KS058, exhibited decreased adhesion to abiotic surfaces under control conditions in comparison to the WT strain. The presence of SU significantly decreased the adhesion of the *fen1Δ/Δ*;*fen12Δ* strain. The addition of CA reduced adhesion of the strain to a limited extent.

In contrast to the *C. albicans* strains with the deletion of the *ERG11* gene or with amino acid substitutions in Erg11p (Figure 9A), the presence of CA did not reduce the adhesion of the *C. albicans* strains with deletions of one or more of the genes involved in SLs biosynthesis, except for the *fen1Δ/Δ*;*fen12Δ/Δ* strain (Figure 9B).

### 2.3. The Combination of Posaconazole and Surfactin or Capric Acid Results in Plasma Membrane Permeabilization in Candida albicans Strains with Altered Ergosterol or Sphingolipids Biosynthesis

Figure 10 presents representative microphotographs of *C. albicans* strains with deletions of the *ERG11* gene, or with amino acid substitutions in Erg11p cultures stained with propidium iodide (PI).

The treatment of all investigated *C. albicans* strains with PSZ resulted in the formation of cell clumps (Figure 10). This effect was heightened in the case of the *C. albicans* SC5314 (WT) strain after the addition of SU to the culture. We observed that most of those cells lost their integrity, which is associated with the increased PM permeabilization. An interesting observation is that the *C. albicans* 10C1B1I1 strain (*ERG11^K143R^*), after treatment with PSZ in combination with SU, formed filaments, which could be a stress response to the presence of SU in the culture medium.

In the case of the *C. albicans* KS058 strain (*erg11Δ/Δ*) treated with a combination of PSZ and CA, we observed cells with disrupted PM integrity. It is worth mentioning that, during the preparation of the staining, KS058 cells treated with PSZ and CA exhibited a reduced capacity for adhesion during centrifugation.

Figure 11 presents representative microphotographs of *C. albicans* strains with deletions of genes involved in SLs biosynthesis cultures stained with propidium iodide (PI).

The treatment of *C. albicans* SN95 (WT), *fen12Δ/Δ* and *fen1Δ/Δ;fen12Δ/Δ* with PSZ alone resulted in the formation of cell clumps, similar to those observed in the case of *C. albicans* strains with alterations in ergosterol biosynthesis (Figure 10). In contrast, the *fen12Δ/Δ* strain exhibited resistance to all combinations of antifungal agents, suggesting that those compounds did not affect the integrity of the PM.

In this group of investigated strains, only *C. albicans* SN95 (WT) produced filaments after treatment with a combination of PSZ and SU (Figure 11). The *C. albicans fen12Δ/Δ* strain treated with PSZ alone, or PSZ in combination with SU, exhibited increased permeabilization of the PM, but not when CA was added to the culture medium.

The *C. albicans fen1Δ/Δ;fen12Δ/Δ* exhibited a tendency to form cell clumps in the presence of PSZ alone, or PSZ in combination with SU or CA, and the permeabilization of the PM was increased in all cases of treated cells in comparison to control conditions.

The summary of the percentages of cells with a permeabilized PM after treatment with PSZ alone, and with a combination of PSZ with SU or CA, is presented in Table 1.

According to these calculations, the most significant increase in the percentage of permeabilization of the PM of cells occurred with a combination of PSZ and SU, compared to control conditions. The only exception to this was the *fen1Δ/Δ* strain, which displayed resistance to all combinations of compounds at the tested concentrations. In comparison to cultures supplemented with PSZ alone, only the SC5314 strain treated with PSZ and SU exhibited an elevated level of PM permeabilization. This suggests that the presence of SU affects *C. albicans* cells predominantly through a reduction in adhesion, rather than by disrupting the PM integrity.

The combination of PSZ and CA did not induce the increased permeabilization of the PM in most of the investigated strains of *C. albicans* when compared to cells treated with PSZ alone. The most vulnerable strains to this treatment were the *ERG11* gene deletant KS058 and a double deletant of the *FEN1* and *FEN12* genes. This result is also supported by the decreased growth and adhesion of those strains in the presence of PSZ and CA (Figure 7, Figure 8 and Figure 9).

We observed an altered morphology of the cells after treatment with a combination of PSZ and SU (different shapes of cells, elongation of cells, formation of cell clumps), but this did not always correspond with a significantly increased permeability of the PM. In the case of the *fen12Δ/Δ* strain cultured with PSZ alone, and PSZ in combination with CA, we observed a mixed culture of intact cells and cells with disrupted PM integrity. The percent of permeabilized cells increased compared to under control conditions, but this change was not statistically significant.

## 3. Discussion

Discovering natural compounds that exhibit potential in azole combination therapy against fungal pathogens poses a challenge for modern medicine. One example of such compound is surfactin (SU), which is examined in this study. Tested concentrations of SU (4–32 µg/mL) did not inhibit growth in any of the tested *C. albicans* strains as an isolated treatment, with the exception of strain KS058 (*erg11Δ/Δ*) at 16 and 32 µg/mL of SU. There are several other azole potentiators that lead to the decreased growth of *Candida* spp. only when in combination with azoles, including 1,4-benzodiazepines and pitavastatin [18,19]. Even supplementing fungal cultures with copper can lead to increased FLC activity, because azoles form a complexes with copper [20]. On the other hand, most of the known azole potentiators exhibit fungistatic effects of their own (e.g., catechol, propolis and derivatives of berberine) [21]. Nevertheless, it is worth investigating the mechanisms of action of these kinds of combinations on fungal cells.

In this study, we investigated whether changes in the *C. albicans* plasma membrane’s (PM) sterol and sphingolipids (SLs) content would result in altered susceptibility to combination treatment with posaconazole (PSZ) and either SU or capric acid (CA). We have previously reported that SU and CA could be promising additives to conventional antifungals such as azoles [11,12,15,16]. We used *C. albicans* strains with amino acid substitutions in Erg11p, which have previously been observed in azole-resistant *Candida* spp. [22,23]. We also employed the *C. albicans erg11Δ/Δ* (KS058) strain, which lacks ergosterol in the PM. Another group of strains key to our investigation consisted of *C. albicans* strains with the deletion of one or more genes involved in SLs biosynthesis. We elected to test combinations of PSZ with SU and CA on these specific *C. albicans* mutants due to the fact that sterols and SLs biosynthesis functionally interact, which can impact the physiology of the PM and fungal cells [24].

Our study indicates that a double substitution in the *ERG11^Y132F,F145L^* in *C. albicans* leads to increased resistance to PSZ, compared to the WT strain (Figure 3). An increased concentration of SU resulted in a successive decrease in growth in all strains except for *C. albicans* KS058. This is due to the lack of Erg11p, which makes this strain resistant to PSZ. Only the highest concentration of SU (32 μg/mL) resulted in the complete growth inhibition of this strain. On the other hand, the *C. albicans fen1Δ/Δ* and *fen12Δ/Δ* strains exhibited increased resistance to PSZ alone at higher concentrations (0.0156–0.125 μg/mL), and to PSZ in combination with SU when compared to the WT strain (Figure 4). The *FEN1* and *FEN12* genes are involved in the elongation of the fatty acids of SLs in fungi; therefore, the deletion of those genes could affect PM structure, thereby reducing the effect of a combination treatment of PSZ and SU on cells. Moreover, *C. albicans fen1Δ/Δ* was found to have an elevated level of ergosterol, which could contribute to its decreased susceptibility to azoles [21].

A combination of PSZ (0.0156–0.125 μg/mL) and an increasing concentration of CA resulted in the decreased growth of *C. albicans* strains 10C1B1I1, 27A5A33A and 9B4B34A compared to the WT strain (Figure 5). This proves that CA induces the synergistic activity with not only fluconazole, but potentially also other azole drugs (e.g., PSZ) [15]. Similar observations were made in relation to the *C. albicans* strains with the deletion of one or more of the genes involved in SLs biosynthesis, despite their higher tolerance to PSZ (Figure 6). The addition of CA successively decreases the growth of those strains, and a combination of PSZ and CA at respective concentrations of 0.0078 μg/mL and 90.6 μg/mL corresponds with significantly reduced growth when compared to the WT strain. For the *C. albicans* strain *fen1Δ/Δ*;*fen12Δ/Δ*, treatment with 45.30 μg/mL CA resulted in total growth inhibition.

Interestingly, a combination treatment of PSZ and SU resulted in a significant reduction in the growth of *C. albicans* strains SC5314 (WT), 10C1B1I1, 27A5A33A and 9B4B34A (Figure 7). We can assume that this is the result of synergistic activity between these two compounds, because in the case of a *C. albicans* strain lacking Erg11p (target for azoles), we observed a decrease in growth rate, but the characteristic growth curve was maintained. The addition of both PSZ with CA resulted in a shift in the timing of the logarithmic phase of growth for all tested strains, with the exception of strain 9B4B34A, but they reached similar OD_600_ at the stationary phase of growth with or without the treatment. Based on the collected data, we can assume that PSZ in combination with SU or CA has a fungistatic character, but that the effect of SU in enhancing this activity is more dramatic than that of CA.

The growth curves determined for *C. albicans* strains with deletions of genes involved in SLs biosynthesis reveal that *fen1Δ/Δ*, *fen12Δ/Δ* and *fen1Δ/Δ;fen12Δ/Δ* strains are more resistant to a combination of PSZ and SU, while strain SN95 (WT) exhibited no growth in the presence of these compounds (Figure 8). The growth curves for these strains are notably shifted in comparison to that of SN95, with a prolonged lag phase, a shifted logarithmic phase regarding control condition and a short stationary phase. The decrease in OD_600_ in the late stages of incubation suggests the eventual death of cells treated with PSZ and SU. On the other hand, similar to *C. albicans* strains with altered sterols biosynthesis (Figure 7), a combination treatment with PSZ and CA corresponds with a shift in the logarithmic phase of growth of the investigated strains, which suggests fungistatic activity. The dramatic growth inhibition in the presence of PSZ and SU, which we recorded for strains SC5314, 10C1B1I1 and 27A5A33A (Figure 8) does not fully correspond with the initial data obtained during susceptibility testing (Figure 1). We assume that this is due to the agitation of cultures during growth curve monitoring, while for susceptibility tests the cultures were incubated stationary.

SU, due to its amphiphilic structure, is known for reducing adhesion among pathogenic bacteria and fungi [25,26]. Fatty acids with short alkyl chains such as CA also exhibit the ability to reduce adhesion and biofilm formation, and to kill pathogenic fungi [12,27]. Moreover, the treatment of *C. albicans* with CA results in the decreased expression of a gene encoding key proteins involved in the adhesion of Hwp1p [16]. Our findings confirm that SU or CA alone can lead to a significant decrease in the adhesion of *C. albicans* with altered sterols biosynthesis (Figure 9A). Taking into consideration that both SU and CA also reduce biofilm formation, this has broad applications for preventing these processes on abiotic surfaces (e.g., medical devices) [28,29].

Our investigations also indicate that combined therapy with PSZ and SU results in PM permeabilization in a broad range of *C. albicans* strains (Table 1), while simultaneously, culturing with PSZ and CA did not correspond with an increased level of PM permeabilization. The addition of CA to cultures did not result in greater PM permeabilization in comparison to cultures treated with PSZ alone, with the exception of the KS058 strain. In relation to PM permeabilization, we can assume that the combination of PSZ and CA has an antagonistic effect. On the other hand, the microscopic analysis revealed that the combination of PSZ and SU led to the formation of cell clumps. This was particularly pronounced in *C. albicans* strains 27A5A33A and 9B4B34A (Figure 10). Moreover, the addition of SU to cultures of *C. albicans* 10C1B1I1 culture resulted in filaments formation, which may be a stress response to the presence of this compound.

The synergistic activity of PSZ with SU or CA can be associated with a reduction in adhesion, which corresponds to interrupted cell proliferation and division. Moreover, the altered cell morphology linked to PM perturbations can lead to a significant reduction in biofilm formation, and, ultimately, a fungistatic effect. Therefore, SU and CA are promising compounds that could be used in combined therapy with azoles to combat fungal pathogens.

## 4. Materials and Methods

### 4.1. Chemicals

Chemicals and reagents were purchased from the following suppliers: posaconazole (PSZ), capric acid (CA), PBS tablets and SDS were purchased from Merck (Darmstadt, Germany); surfactin (SU) was purchased from Kaneka Corporation (Tokyo, Japan); propidium iodide (PI) and D-glucose were purchased from Bioshop (Burlington, ON, Canada); peptone, yeast extract and HCl were purchased from Thermo Fisher Scientific (Waltham, MA, USA); isopropanol was purchased from Chempur (Piekary Śląskie, Poland); sodium chloride (NaCl) was purchased from StanLAB (Lublin, Poland). All chemicals were of a high purity grade.

### 4.2. Strains and Growth Conditions

The *C. albicans* strains used in this study are listed in Table 2.

Strains were pre-cultured in YPD medium (1% YE, 1% peptone, 2% glucose) for 24 h, at a temperature of 28 °C with shaking at 120 rpm.

### 4.3. C. albicans Susceptibility Testing

In order to determine the MIC values, a serial dilution of PSZ (0–0.125 µg/mL; original PSZ stock 1 mg/mL in DMSO) alone or in combination with SU (0–32 µg/mL; original SU stock 1 mg/mL in H_2_O_dd_) or with capric acid (0–90.6 µg/mL; original CA stock 45.3 mg/mL in methanol) was prepared in YPD medium using sterile 96-well plates (Sarstedt, Nümbrecht, Germany) [17]. Then, the YPD medium was inoculated with *C. albicans* strains at a final OD_600_ = 0.01 and cultured overnight (28 °C, stationary). After this time, the OD_600_ was measured using a plate reader (ASYS UVM 340 Biogenet, Cosenza, Italy). The negative control consisted of a serial dilution of the compounds in the same media, without *C. albicans* inoculation. The positive control (100% of growth) consisted of *C. albicans* cultures in YPD medium without the tested compounds. The experiment was performed in 3 biological repetitions.

### 4.4. C. albicans Growth Curves

To determine the growth phases in the presence of PSZ alone or in combination with SU or CA, the *C. albicans* strains were pre-grown in the YPD medium (28 °C; with shaking at 120 rpm) overnight. Then 100 μL of fresh YPD medium was added to the sterile 96-well plate (Sarstedt, Germany) and inoculated with *C. albicans* to the final OD_600_ = 0.1. The OD_600_ was registered for 24 h (or 48 h if indicated) in 1 h intervals using the Spark multimode microplate reader (Tecan, Männedorf, Switzerland) as described before [31]. The experiment was performed with 3 biological repetitions.

### 4.5. Adhesion of C. albicans to Polystyrene

SU and CA were assessed as *C. albicans* adhesion inhibitors using 96-well plates (Sarstedt, Germany) using a previously described method [34]. Firstly, the wells were preincubated with 100 µL of SU (4 µg/mL) and CA (45.3; 22.7 or 11.35 µg/mL, depending on the strain used) in a solution of PBS for 2 h at 37 °C (stationary). PBS buffer was used as a positive control. Next, the solutions were removed from the plate, and the wells were rinsed three times with PBS buffer. *C. albicans* suspensions in PBS were diluted to final OD_600_ = 0.6. The prepared inoculum was transferred to a 96-well plate (100 µL) and incubated under the same conditions (2 h, 37 °C, stationary). After this time, all supernatants were removed, and the wells were washed three times with PBS to remove nonadherent cells. The remaining adhered cells were stained with 0.1% crystal violet solution (5 min, 25 °C), and then the wells were washed three times with fresh PBS. Then the PBS was removed, and dye was released by adding 200 µL of 0.05M HCl with 1% SDS to the isopropanol. Absorbance at 590 nm was registered using a plate reader (ASYS UVM 340 Biogenet, Cosenza, Italy). This experiment was performed in 6 repetitions.

### 4.6. Analysis of Plasma Membrane Permeabilization

To assess the effects of PSZ alone or in combination with SU or CA on PM permeability, we performed the propidium iodide (PI) staining of cells [17]. *C. albicans* cultures treated for 24 h (YPD, 20 mL, 28 °C, 120 rpm) without tested compounds, with PSZ (0.0156 µg/mL), with PSZ (0.0078 or 0.0156 µg/mL, depending on the used strain) and SU (4 µg/mL), or with PSZ (0.0156 µg/mL) and CA (45.3; 22.7 or 11.35 µg/mL, depending on strain used), were centrifuged (5 min, 4500 rpm), washed twice with 0.9% NaCl and resuspended in fresh 0.9% NaCl, then adjusted to OD_600_ = 0.3. The cells were stained with propidium iodide (4 μg/mL) and incubated at 25 °C for 5 min, in the dark, then washed twice with 0.9% NaCl (5 min, 4500 rpm), concentrated, and observed under a Zeiss Axio Imager A2 microscope equipped with a Zeiss Axiocam 503 mono microscope camera and a Zeiss HBO100 mercury lamp (Poznań, Poland). The percentage of permeabilization was evaluated by calculating PI-positive cells as a proportion of at least 150 cells in three independent replicates for each experiment (described as a population of cells).

### 4.7. Statistical Analysis

For data analysis, statistical significance was determined using a Student’s *t*-test (binomial, unpaired). Data represent the means ± standard deviations (SD) of at least three biological replicates. In the case of the data presented in Table 1, the mean percentage of permeabilization was determined from the population (consisting of 3 independent biological replicates) and the standard error (SE) was determined according to the following formula:$$SE=\sqrt{\frac{p\times q}{n}}$$
where p = percent (%) of cells with permeabilized PM in the population; q = percent (%) of cells without permeabilized PM in the population; n = number of cells in population.

The data presented in Figure 3, Figure 4, Figure 5 and Figure 6 and Figure 9 were prepared using GraphPad Prism 6.0. (GraphPad Software, San Diego, CA, USA).

## 5. Conclusions

The main conclusions of our research are as follows:PSZ in combination with SU and CA leads to the decreased growth of *C. albicans*. The level of growth inhibition is dependent on the types of alterations in the ergosterol or SLs biosynthesis pathways of *C. albicans* strains;The strain most susceptible to the combination of PSZ and SU is *C. albicans* that lacks the *ERG11* gene. The most susceptible to the combination of PSZ and CA is the strain of *C. albicans* that lacks both *FEN1* and *FEN12* genes;PSZ in combination with SU causes the significantly decreased adhesion of all investigated *C. albicans* strains. The combination of PSZ with CA does not lead to the impaired adhesion of *C. albicans* with the deletion of genes involved in SLs biosynthesis, and for *C. albicans ERG11* gene mutants, the effect is less dramatic comparing to a combined therapy of PSZ and SU;The combination of PSZ with SU and CA does not correspond with increased *C. albicans* PM permeabilization compared to control conditions. Nevertheless, the formation of cell clumps was observed due to the use of PSZ with SU and CA. Therefore, the synergy between PSZ and SU or CA is based on a reduction in the adhesion of *C. albicans*, rather than PM permeabilization.

## Figures and Tables

**Figure 1 ijms-24-17499-f001:**
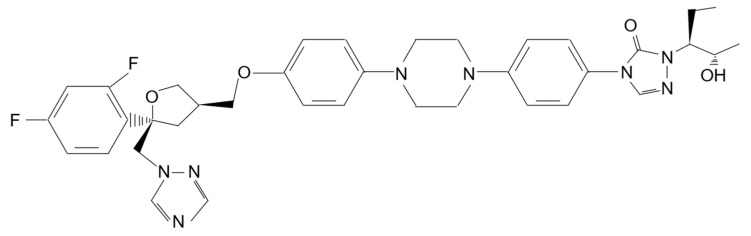
Chemical structure of posaconazole (PSZ).

**Figure 2 ijms-24-17499-f002:**
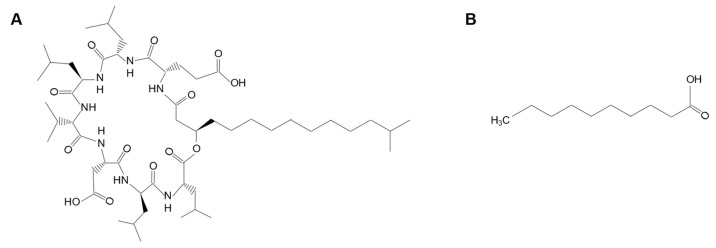
Chemical structure of surfactin (SU (**A**)) and capric acid (CA (**B**)).

**Figure 3 ijms-24-17499-f003:**
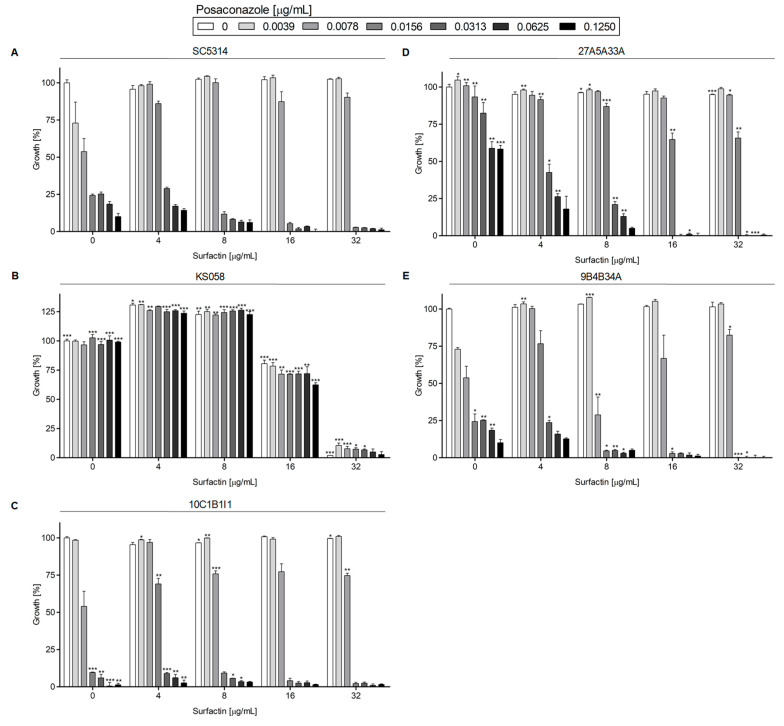
Growth (%) of *C. albicans* strains with the deletion of *ERG11* gene or with amino acid substitutions in Erg11p in the presence of PSZ (0–0.125 µg/mL) with or without the addition of SU (0–32 µg/mL). The *C. albicans* SC5314 (WT (**A**)), KS058 (*erg11Δ/Δ* (**B**)), 10C1B1I1 (*ERG11^K143R^* (**C**)), 27A5A33A (*ERG11^Y132F,F145L^* (**D**)) and 9B4B34A (*ERG11^Y132F,K143R^* (**E**)) strains were cultured stationary for 24 h at a temperature of 28 °C. The experiment was performed in 3 replicates and obtained data were analyzed using Student’s t-test (*, *p* < 0.05; **, *p* < 0.01; ***, *p* < 0.001). The data for *C. albicans* KS058, 10C1B1I1, 27A5A33A and 9B4B34A strains were compared to those obtained for *C. albicans* SC5314 (WT) in certain concentrations of PSZ or SU.

**Figure 4 ijms-24-17499-f004:**
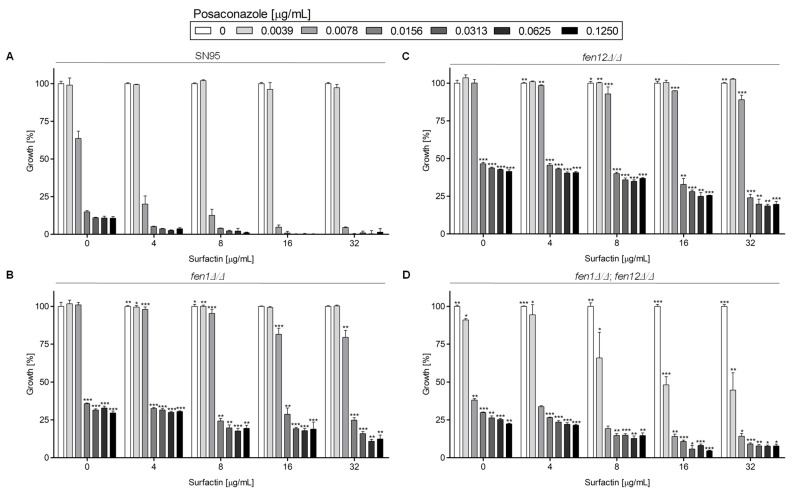
Growth (%) of *C. albicans* strains with deletion of genes involved in sphingolipids (SLs) biosynthesis in the presence of PSZ (0–0.125 µg/mL) with or without addition of SU (0–32 µg/mL). The *C. albicans* SN95 (WT (**A**)), *fen1Δ/Δ (***B**), *fen12Δ/Δ* (**C**) and *fen1Δ/Δ*;*fen12Δ/Δ* (**D**) strains were cultured stationary for 24 h at a temperature of 28 °C. The experiment was performed in 3 replicates and obtained data were analyzed using Student’s t-test (*, *p* < 0.05; **, *p* < 0.01; ***, *p* < 0.001). The data for *C. albicans fen1Δ/Δ*, *fen12Δ/Δ* and *fen1Δ/Δ* and *fen12Δ/Δ* strains were compared to those obtained for *C. albicans* SN95 (WT) in certain concentrations of PSZ or SU.

**Figure 5 ijms-24-17499-f005:**
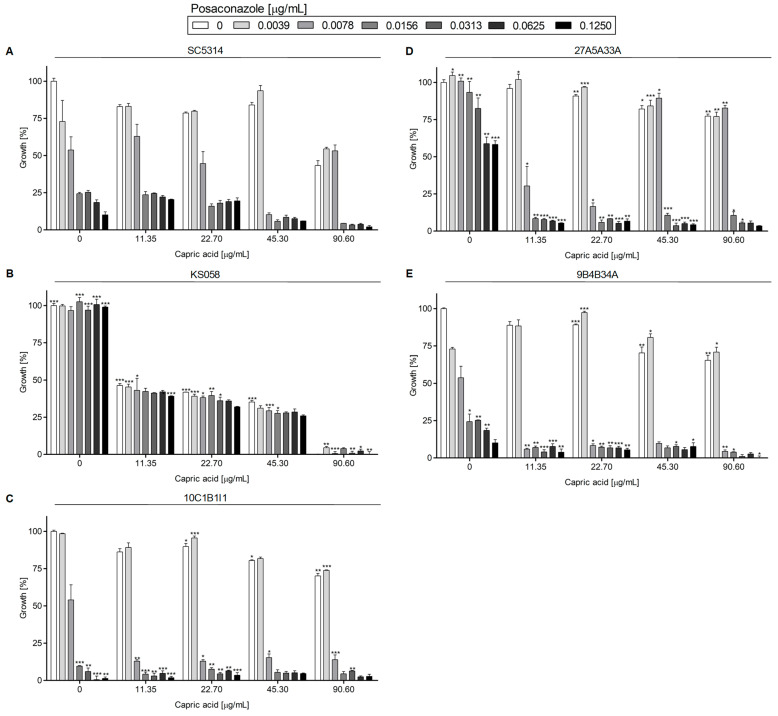
Growth (%) of *C. albicans* strains with the deletion of the *ERG11* gene or with amino acid substitutions in Erg11p in the presence of PSZ (0–0.125 µg/mL) with or without the addition of capric acid (0–90.60 µg/mL). The *C. albicans* SC5314 (WT (**A**)), KS058 (*erg11Δ/Δ* (**B**)), 10C1B1I1 (*ERG11^K143R^* (**C**)), 27A5A33A (*ERG11^Y132F,F145L^* (**D**)) and 9B4B34A (*ERG11^Y132F,K143R^* (**E**)) strains were cultured stationary for 24 h at a temperature of 28 °C. The experiment was performed in 3 replicates and obtained data were analyzed using Student’s t-test (*, *p* < 0.05; **, *p* < 0.01; ***, *p* < 0.001). The data for *C. albicans* KS058, 10C1B1I1, 27A5A33A and 9B4B34A strains were compared to those obtained for *C. albicans* SC5314 (WT) in certain concentrations of PSZ or SU.

**Figure 6 ijms-24-17499-f006:**
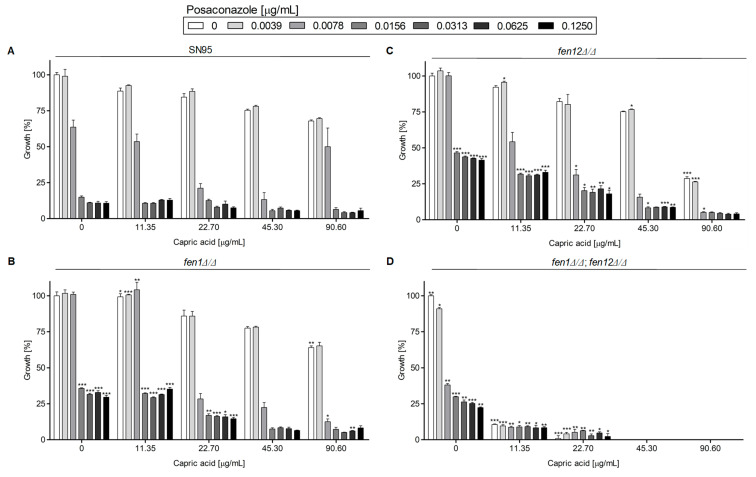
Growth (%) of *C. albicans* strains with the deletion of genes involved in sphingolipids (SLs) biosynthesis in the presence of PSZ (0–0.125 µg/mL) with or without the addition of capric acid (0–90.60 µg/mL). The *C. albicans* SN95 (WT (**A**)), *fen1Δ/Δ* (**B**), *fen12Δ/Δ* (**C**) and *fen1Δ/Δ*;*fen12Δ/Δ* (**D**) strains were cultured stationary for 24 h at a temperature of 28 °C. The experiment was performed in 3 replicates and obtained data were analyzed using Student’s t-test (*, *p* < 0.05; **, *p* < 0.01; ***, *p* < 0.001). The data for *C. albicans fen1Δ/Δ*, *fen12Δ/Δ* and *fen1Δ/Δ;fen12Δ/Δ* strains were compared to those obtained for *C. albicans* SN95 (WT) in certain concentrations of PSZ or SU.

**Figure 7 ijms-24-17499-f007:**
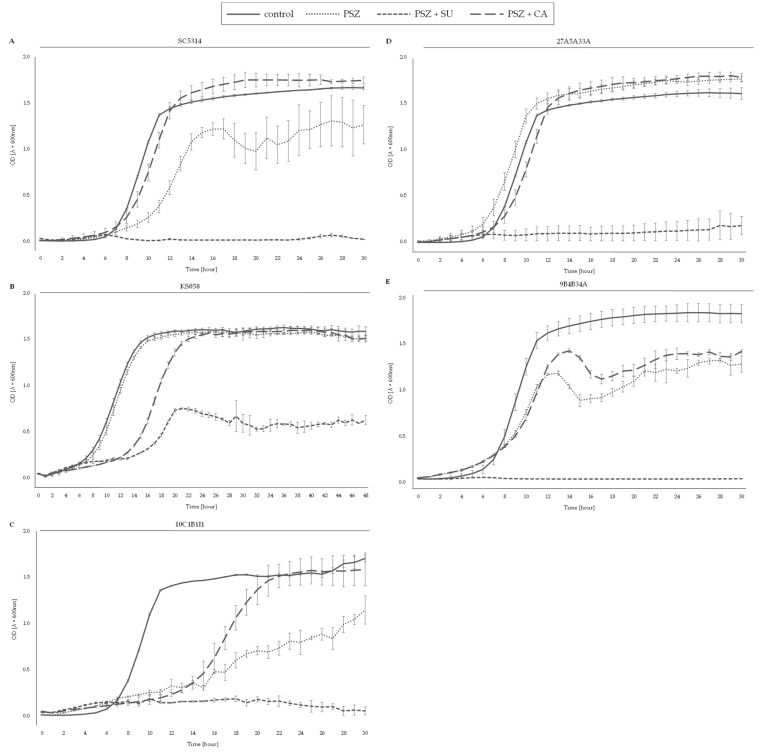
Growth curves of *C. albicans* strains with deletion of *ERG11* gene or with amino acid substitutions in Erg11p under control conditions (YPD), in the presence of PSZ (0.0156 µg/mL) with or without the addition of surfactin (4 µg/mL; PSZ = 0.0156 µg/mL) or capric acid (45.3 µg/mL; PSZ = 0.0078 µg/mL). The *C. albicans* SC5314 (WT (**A**)), KS058 (*erg11Δ/Δ* (**B**)), 10C1B1I1 (*ERG11^K143R^* (**C**)), 27A5A33A (*ERG11^Y132F,F145L^* (**D**)) and 9B4B34A (*ERG11^Y132F,K143R^* (**E**)) strains were cultured for 24 h (or 48 h in the case of KS058 strain) at a temperature of 28 °C, with shaking (120 rpm). The experiment was performed in 3 replicates and the figure represents mean absorbance (λ = 600 nm) ± SD.

**Figure 8 ijms-24-17499-f008:**
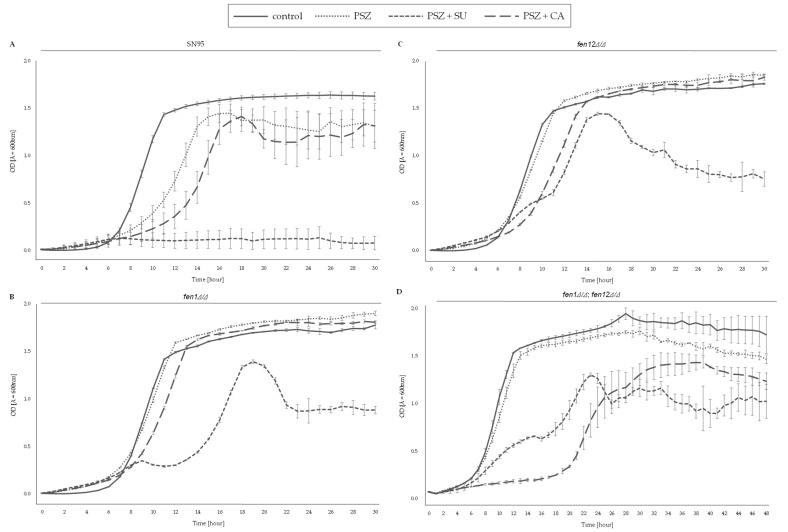
Growth curves of *C. albicans* strains with deletion of genes involved in sphingolipids (SLs) biosynthesis in the presence of PSZ (0.0156 µg/mL) with or without the addition of surfactin (4 µg/mL) or capric acid (22.7 µg/mL or for *fen1Δ/Δ* and *fen12Δ/Δ* strain 11.35 µg/mL). The *C. albicans* SN95 (WT *(***A**)), *fen1Δ/Δ* (**B**), *fen12Δ/Δ* (**C**) and *fen1Δ/Δ*;*fen12Δ/Δ* (**D**) strains were cultured for 24 h (or 48 h in case of *fen1Δ/Δ* and *fen12Δ/Δ* strain) at a temperature of 28 °C, with shaking (120 rpm). The experiment was performed in 3 replicates and the figure represents mean absorbance (λ = 600 nm) ± SD.

**Figure 9 ijms-24-17499-f009:**
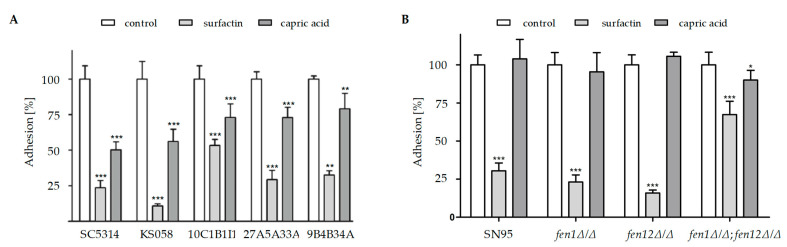
The adhesion (%) of *C. albicans* strains with altered ergosterol (SC5314 (WT), KS058, 10C1B1I1, 27A5A33A and 9B4B34A (**A**)) or sphingolipids biosynthesis (SN95 (WT), *fen1Δ/Δ*, *fen12Δ/Δ* and *fen1Δ/Δ*;*fen12Δ/Δ*) to abiotic surface under control conditions (PBS) or in the presence of surfactin (SU, 4 μg/mL) or capric acid (CA = 45.3 μg/mL for strains at (**A**); 22.7 μg/mL for strains at (**B**) or 11.35 μg/mL for *C. albicans fen1Δ/Δ*;*fen12Δ/Δ* strain). The data obtained for *C. albicans* strains incubated with SU or CA were compared to those under control conditions. The experiment was performed in 6 replicates and obtained data were analyzed using Student’s t-test (*, *p* < 0.05; **, *p* < 0.01; ***, *p* < 0.001).

**Figure 10 ijms-24-17499-f010:**
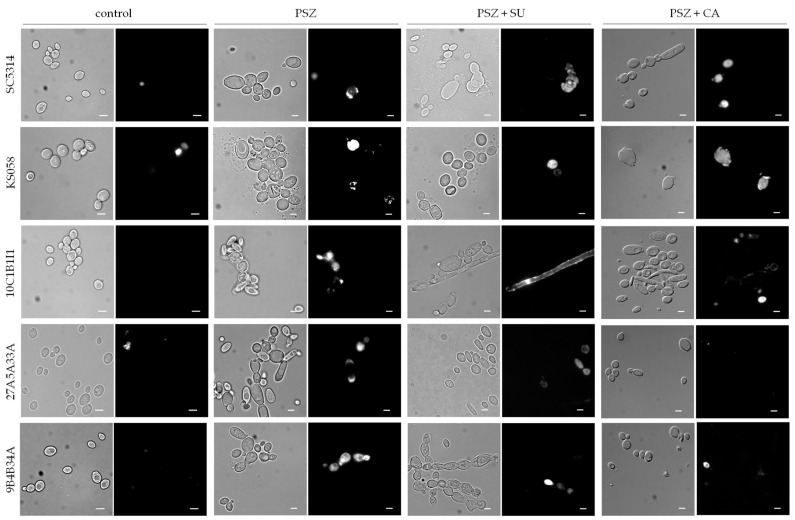
The permeabilization of the plasma membrane (PM) of *C. albicans* strains SC5314 (WT), KS058, 10C1B1I1, 27A5A33A and 9B4B34A after 24 h of culturing with or without the presence of PSZ (0.0156 μg/mL), PSZ with SU (0.0156 with 4 μg/mL, respectively), or PSZ with CA (0.0078 with 45.3 μg/mL, respectively). The cells were stained with propidium iodide (PI, 4 μg/mL) for 5 min and then analyzed for PM permeabilization. The figure contains the representative microphotographs (scalebar = 50 µm). For every strain under each condition, at least 150 cells were analyzed.

**Figure 11 ijms-24-17499-f011:**
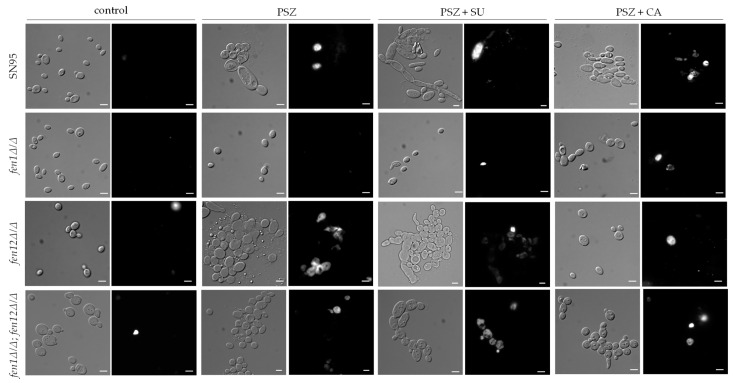
The permeabilization of the plasma membrane (PM) of *C. albicans* strains SN95 (WT), *fen1Δ/Δ*, *fen12Δ/Δ* and *fen1Δ/Δ*;*fen12Δ/Δ* after 24 h culture with or without the presence of PSZ (0.0156 μg/mL), PSZ with SU (0.0156 with 4 μg/mL, respectively) or PSZ with CA (0.0156 with 22.7 μg/mL or 0.0156 μg/mL with 11.35 μg/mL for *fen1Δ/Δ* and *fen12Δ/Δ* strain, respectively). The cells were stained with propidium iodide (PI, 4 μg/mL) for 5 min and then analyzed for PM permeabilization. The figure contains representative microphotographs (scalebar = 50 µm). For every strain under each condition, at least 150 cells were analyzed.

**Table 1 ijms-24-17499-t001:** The permeabilization (%) of the plasma membrane (PM) of *C. albicans* strains with altered ergosterol (SC5314 (WT), KS058, 10C1B1I1, 27A5A33A and 9B4B34A) or sphingolipids biosynthesis (SN95 (WT), *fen1Δ/Δ*, *fen12Δ/Δ* and *fen1Δ/Δ;fen12Δ/Δ*) pathways. The cells were stained with propidium iodide (PI, 4 μg/mL) for 5 min and then analyzed for PM permeabilization. For every strain under each condition, at least 150 cells were analyzed in three independent biological repetitions. The data represent mean permeabilization (%) ± SE from the population (*, *p* < 0.05; **, *p* < 0.01). Obtained data were analyzed using Student’s t-test.

*C. albicans* Strain	Control	PSZ	PSZ + SU	PSZ + CA
SC5314	0.55 ± 0.32	0.55 ± 0.32	0.55 ± 0.32	0.55 ± 0.32
KS058	5.78 ± 1.10	5.78 ± 1.10	5.78 ± 1.10	5.78 ± 1.10
10C1B1I1	33.76 ± 2.67 *	33.76 ± 2.67 *	33.76 ± 2.67 *	33.76 ± 2.67 *
27A5A33A	10.04 ± 1.30	10.04 ± 1.30	10.04 ± 1.30	10.04 ± 1.30
9B4B34A	3.10 ± 1.08	3.10 ± 1.08	3.10 ± 1.08	3.10 ± 1.08
SN95	14.55 ± 2.13 **	14.55 ± 2.13 **	14.55 ± 2.13 **	14.55 ± 2.13 **
*fen1Δ/Δ*	12.57 ± 1.73 **	12.57 ± 1.73 **	12.57 ± 1.73 **	12.57 ± 1.73 **
*fen12Δ/Δ*	36.43 ± 4.24 *	36.43 ± 4.24 *	36.43 ± 4.24 *	36.43 ± 4.24 *
*fen1Δ/Δ*; *fen12Δ/Δ*	4.85 ± 1.22	4.85 ± 1.22	4.85 ± 1.22	4.85 ± 1.22

**Table 2 ijms-24-17499-t002:** *C. albicans* strains used in this study.

Strain	Genotype	References
SC5314	*URA3/URA3* (clinical isolate)	Prof. D. Sanglard [30]
KS058	*erg11Δ::SAT1-FLIP/erg11Δ::FRT*	[31]
10C1B1I1	*ERG11^K143R^::FRT/ERG11^K143R^::FRT*	Prof. D. Rogers [32]
27A5A33A	*ERG11^Y132F,F145L^::FRT/ERG11^Y132F,F145L^::FRT*	
9B4B34A	*ERG11^Y132F,K143R^::FRT/ERG11^Y132F,K143R^::FRT*	
SN95	*arg4Δ/arg4Δ his1Δ/his1Δ URA3/ura3::imm434 IRO1/iro1::imm434*	Prof. K. Ganesan [33]
*fen1* *Δ* */* *Δ*	as SN95 but *fen1Δ/fen1Δ*	
*fen12* *Δ* */* *Δ*	as SN95 but *fen12Δ/fen12Δ*	
*fen1* *Δ* */* *Δ* *;fen12* *Δ* */* *Δ*	as SN95 but *fen1Δ/fen1Δ;fen12Δ/fen12Δ*	

## Data Availability

Data is contained within the article and Appendix A.

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
