# Peer review of "Surfactin and Capric Acid Affect the Posaconazole Susceptibility of Candida albicans Strains with Altered Sterols and Sphingolipids Biosynthesis"

_ijms, 2023, doi:10.3390/ijms242417499_

Round 1

Reviewer 1 Report

Comments and Suggestions for Authors

This study presents findings on the synergistic effects of posaconazole in combination with surfactin and capric acid against Candida albicans strains with altered ergosterol and sphingolipid properties. Since resistant C. albicans strains are global concern, these results are important. However, the presentation of the study is unclear because of the deficient explanations regarding the rationale and methodology. To enhance clarity, here is my comments and suggestions.

-        - Minor revisions needed in English

-       - Line 60-65. Improve this paragraph. Which strains were used in this study and why? Which studies were carried out and why? Explain the purpose and plan more clearly and understandably.

-          Line 66-81. It is confusing that the study results are in the introduction. These paragraphs are better suited to conclusion or discussion.

-          What is your significance level in statistical analysis? Write the meanings of the stars in the figures under the figures according to their significance levels.

-          Surfactin and capric acid have antifungal effects on their own. In this case, were the MIC values of these substances checked? Were the MIC values of only posiconazole determined against the strains? Has the combination MIC been determined? Or were only growth percentages determined? If there are MIC values, make a table and write these values. If not done, Line 411 in the material method is misleading. Change the title.

-          Include the characteristics of the strains in Table 2. What kind of mutants are these? Which genes are missing in detail etc.

-          Line 453. What does the PF abbreviation refer?

Comments on the Quality of English Language

Minor English corrections are needed. Check and correct the punctuations and conjunctions

Author Response

  1. Line 60-65. Improve this paragraph. Which strains were used in this study and why? Which studies were carried out and why? Explain the purpose and plan more clearly and understandably.

Thank you for this comment. This work is the continuation of our previous study where we determined the synergistic activity of fluconazole and surfactin or capric acid against C. albicans. Here we wanted to determine if this effect is specific for fluconazole or we will observe similar effect for posaconazole which has a different chemical structure comparing to fluconazole. We also chose C. albicans strains with altered ergosterol and sphingolipids (SLs) level because we wanted to determine whether those changes will affect the susceptibility to used combined therapy. In our strain collection, we have SLs mutants (fen1D/D, fen12D/D and fen1D/Dfen12D/D) constructed from C. albicans SN95 strain. The parental strain for C. albicans SN95 is SC5314 strain. Similarly, we possess the strains with alterations in ERG11 gene constructed only from C. albicans SC5314 parental strain. Therefore, we decided to use such selection of strains in this study.

According to your comment we decided to add the explanation of our studies in lines 82 – 86.

  1. Line 66-81. It is confusing that the study results are in the introduction. These paragraphs are better suited to conclusion or discussion.

Thank you for that comment. We decided to delete fragment of introduction which partially described obtained results. We also added the fragment in the end of introduction section which justify and explains the conducted research (lines: 82 - 86).

  1. What is your significance level in statistical analysis? Write the meanings of the stars in the figures under the figures according to their significance levels.

Thank you for this remark. We implemented adequate description in lines: 106 (Figure 3), 131 (Figure 4), 149 - 150 (Figure 5), 164 (Figure 6), 228 (Figure 9) and 297 (Table 1).

  1. Surfactin and capric acid have antifungal effects on their own. In this case, were the MIC values of these substances checked? Were the MIC values of only posaconazole determined against the strains? Has the combination MIC been determined? Or were only growth percentages determined? If there are MIC values, make a table and write these values. If not done, Line 411 in the material method is misleading. Change the title.

Thank you for this remark. The aim of our study was to investigate whether combination of posaconazole with surfactin or capric acid will result in diminished growth of the pathogen. Therefore, all the white bars in figures 1-4 represent the 100% of growth of C. albicans strains in presence of certain concentration of surfactin or capric acid and in the absence of posaconazole. We chose this way of presentation of our data because we were more focused on showing the synergistic activity of tested compounds with posaconazole. In order to present more clearly the antifungal effect of surfactin or capric acid on their own on investigated C. albicans strains we changed the way of presentation of our results in figure 3, figure 4, figure 5, figure 6. Now we present every set of data where surfactin or capric acid were added to culture, as the percent of growth in accordance to the growth of C. albicans without addition of any compound (neither PSZ nor SU or CA). Now we think that the reduction of growth of C. albicans strains are clearer to understand. We also added the additional table where we show the MIC values for every strain in supplementary material.

In case of data for only posaconazole against used strains they are presented in every figure (figures 3-6) described as “0 mg/mL” of surfactin or capric acid.

We also deleted “(determination of minimal inhibitory concentration (MIC) value)” from the Materials and Method section (line 433) according to your suggestion.

  1. Include the characteristics of the strains in Table 2. What kind of mutants are these? Which genes are missing in detail etc.

Thank you for this question. The full genotype characteristic of used strains is present in Table 2 (in section “genotype”). We also added the characteristic of used C. albicans strains in the introduction part (lines: 77 – 81) in order to clarify the the aim of research.

  1. Line 453. What does the PF abbreviation refer?

Thank you for this remark. We already replaced “PF” abbreviation for “0.9% NaCl” in order to clarify the described method (lines 475-476 and 478).

Reviewer 2 Report

Comments and Suggestions for Authors

The current manuscript "Surfactin and capric acid affects the posaconazole susceptibility of Candida albicans strains with altered sterols and sphingolipids biosynthesis" The idea seems to be good and the results are promising for publication. However, some comments and modification have to address before publication.

-          In the abstract, the authors should add quantitative values to better reflect the researchers' findings.

-          The abstract part needs to rewrite in a way to define the exact novelty and originality of your work.

-          All abbreviations used should be mentioned in the place of their first mention followed by an abbreviation and then only the abbreviation is written.

-          The introduction must be completed by clarifying the main objectives of the research and by motivating the experimental strategy adopted by authors.

-          Please specify the impact of your study in the introduction.

-          The authors should add more explanations to the results.

-          The authors need to improve the resolution of all figures.

-          Conclusion??

-          Add appendix for abbreviation.

-          Add references to all methods.

-          The whole manuscript must be checked to avoid the presentation of same information several times.

-          The English language needs to be significantly improved, in wording, grammar and sentence structure.

-          I suggest the authors to go through the manuscript one more time to minimize some errors, typos etc.

Comments on the Quality of English Language

 The English language needs to be significantly improved, in wording, grammar and sentence structure.

Author Response

The current manuscript "Surfactin and capric acid affects the posaconazole susceptibility of Candida albicans strains with altered sterols and sphingolipids biosynthesis" The idea seems to be good and the results are promising for publication. However, some comments and modification have to address before publication.

- In the abstract, the authors should add quantitative values to better reflect the researchers' findings.

Thank you for this remark. We added the quantitative value in abstract (lines: 17-18) to better reflect our findings. We decided to add an information about the significant reduction of adhesion of investigated strains of C. albicans because we think that this aspect of our research is the most important.

- The abstract part needs to rewrite in a way to define the exact novelty and originality of your work.

Thank you for this comment. In order to clarify purpose of our research and to indicate the exact novelty of our work we decided to delete to last part of introduction part and we replaced it with the fragment which directly explain the originality of our research (lines: 86-91). We previously investigated the synergy between fluconazole and surfactin or capric acid only on C. albicans WT and erg11D/D strains. The novelty of our research is an investigation of the synergistic activity of other azole drug (posaconazole) with those additives on broad spectrum of C. albicans strains with alterations in ergosterol or sphingolipids (SLs) biosynthesis pathway. In present work we showed for the first time that the synergy is not specific for a fluconazole. Moreover, deletions of one or more genes involved in sterol or SLs biosynthesis impact the susceptibility of C. albicans to the combined therapy.

- All abbreviations used should be mentioned in the place of their first mention followed by an abbreviation and then only the abbreviation is written.

Thank you for this remark. We double checked all abbreviations and we left the full names only in the place of their first mention (e.g., posaconazole, surfactin, capric acid and sphingolipids) following by an abbreviation in main part of the manuscript. The only exceptions are abstract, titles of chapters and description of figures presented.

- The introduction must be completed by clarifying the main objectives of the research and by motivating the experimental strategy adopted by authors.

Thank you for this remark. The main aim of our study is now emphasized in the introduction part (lines: 76-77). We also deleted the last part of our introduction and replaced it with the clear explanation why the specific C. albicans strains were used and what was the experimental strategy for our study (lines: 79-89). We also indicated that present study is the continuation of our previous work which focused on other type of azole drug (fluconazole, FLC) and here we wanted to investigate whether the synergy with SU or CA is specific to FLC.

- Please specify the impact of your study in the introduction.

Thank you for this remark. We specified the impact of our study at the end of the introduction part (lines: 89-91).

- The authors should add more explanations to the results.

Thank you for that comment. We decided to change the method of presentation of our results in Figures 3-6 and now we hope that explanations for obtained data are more clear. We also made some changes in the discussion part and we also added the fragment about the examples of other azoles potentiators which, in our opinion, enrich the discussion of our result (lines: 331-341). We think that all obtained results are fully discussed and confronted with the literature.

- The authors need to improve the resolution of all figures.

Thank you for this remark. We tried to improve the resolution of all figures and we hope that now the reception of the figures is better.

- Conclusion??

Thank you for this remark. We created conclusion section in lines: 513-530.

- Add appendix for abbreviation.

Thank you for this comment. We added the appendix A (lines: 527-535) for abbreviations.

Add references to all methods.

Thank you for this remark. We added references to all methods which were used during this study (lines: 454, 468-469, 473 and 489).

- The whole manuscript must be checked to avoid the presentation of same information several times.

Thank you for this comment. We double checked manuscript to avoid the presentation of same information several times.

- The English language needs to be significantly improved, in wording, grammar and sentence structure. I suggest the authors to go through the manuscript one more time to minimize some errors, typos etc.

Thank you for this remark. We employed native speaker in order to improve our manuscript. We think that those changes in English language provide better reception of our manuscript.

Reviewer 3 Report

Comments and Suggestions for Authors

Author Response

The publication presented for evaluation is one of a series investigating the interactions of azoles with surfactin and their impact on Candida albicans strains with altered sterols and sphingolipid biosynthesis. Additionally, the study analyzed the influence of capric acid.

1a. The introduction is concise, only lines 25-59.

At the moment, the Introduction is extended and include lines 25 to 91.

1b. The introduction should include a drawing with the structures of the tested compounds. This is important because the compounds contain many functional groups, especially posaconazole and surfactin.

Thank you for this remark. We added the drawings of chemical structure of posaconazole (Figure 1, line: 44), surfactin and capric acid (Figure 2; line: 75).

1c. The introduction should include a justification for the research. Unfortunately, the authors described the results obtained here...

Thank you for that comment. We decided to delete fragment of introduction which partially described obtained results. We also added the fragment in the end of introduction section which justify conducted research (lines: 79–91).

  1. Nine Candida strains were selected for testing, against which surfactin (SU) and capric acid (CA) showed zero growth inhibiting activity. Why weren't strains from the authors' earlier publications, whose growth was inhibited by surfactin, used?

Thank you for this question. This research is the continuation of our previous study on synergistic activity of azoles (here PSZ, we previously investigated fluconazole (FLC)) with SU and CA. Our previous study indicates that altered ergosterol level impacts the susceptibility of C. albicans to such therapy. Taking into account that we previously proved that FLC exhibit the synergistic activity with SU or CA, here we wanted to determine if this effect is specific for FLC or we will observe similar effect for PSZ which has a significantly different chemical structure comparing to FLC.

            We also chose different selection of C. albicans strain because we wanted to determine whether the altered sphingolipids (SLs) level in mutant strains will affect the susceptibility to used combined therapy. In our strain collection, we have SLs mutants (fen1D/D, fen12D/D and fen1D/Dfen12D/D) constructed from C. albicans SN95 strain. The parental strain for C. albicans SN95 is SC5314 strain. Similarly, we possess the strains with alterations in ERG11 gene constructed only from C. albicans SC5314 parental strain. Therefore, we decided to use such selection of strains in this study.

In order to present more clearly the antifungal effect of surfactin or capric acid on their own on investigated C. albicans strains we changed the way of presentation of our results in Figure 3, Figure 4, Figure 5, Figure 6. Now we present every set of data where surfactin or capric acid were added to culture, as the percent of growth in accordance to the growth of C. albicans without addition of any compound (neither PSZ nor SU or CA). Now we think that the reduction of growth of C. albicans strains are clearer to understand.

  1. Why are the data from Fig. 1 and 2 different from the data from the experiments shown in Fig. 5? The growth curves of C. albicans strains in the presence of posaconazole (PSZ) (0.0156

µ g /mL) and PSZ with surfactin (SU (4 µ g /mL) or capric acid (CA) (45.3 µ g /mL) are presented in figure 5 cannot differ significantly from the data from the experiments described in Figures 1 and 2. Let's only analyze the data for one strain, 27A533A. The results for the same strain and posaconazole are different. The MIC 50 read from graph I is above 0.125 µg/mL; in the second experiment (graph II), it is less than 0.0078 µg/mL. In the third experiment (III) at a concentration of 0.0156 µg/mL, the grow th after 24 hours was not inhibited compared to the control.

Pasaconazole (0.0156 µg/mL in the presence of 4 µg/mL surfactin inhibits the growth of Candida albicans 27A533A by approximately 25% (Graph II). In Graph III, at the same concentrations, it inhibits the growth by 100%. We observe an identical situation for capric acid, where in graph II, we observe inhibition of growth up to 12%, and in graph III, there is no inhibition growth. There are also similar inaccuracies in other experiments in this series.

Thank you for this comment. We already applied changes in Figures 3–6 which were described in our answer to your second question. Comparing now the obtained results presented in Figures 3–6 to those obtained in Figures 9 and 10 we created table summarizing these results:

C. albicans strain

PSZ

PSZ + SU

PSZ + CA

MIC (incubated stationary)

Growth curve (shaking)

MIC (inhibited stationary)

Growth curve (shaking)

MIC (inhibited stationary)

Growth curve (shaking)

SC5314

~25% of growth

late log phase, lower OD values

~80% of growth

no growth

~15% of growth

similar growth to control

Comment

Data are consistent

Data are not consistent

Data are not consistent

KS058

100% of growth

similar growth to control

~100% of growth

significantly diminished growth

~30% of growth

late log phase

Comment

Data are consistent

Data are not consistent

Data are consistent

10C1B1I1

10% of growth

late log phase, lower OD values

~80% of growth

no growth

~20% of growth

late log phase

Comment

Data are consistent

Data are not consistent

Data are consistent

27A5A33A

100% of growth

similar growth to control

~100% of growth

no growth

~100% of growth

similar growth to control

Comment

Data are consistent

Data are not consistent

Data are consistent

9B4B34A

~25% of growth

late log phase, lower OD values, characteristic stationary phase

~75% of growth

no growth

~10% of growth

late log phase, lower OD values, characteristic stationary phase

Comment

Data are consistent

Data are not consistent

Data are consistent

SN95

~15% of growth

late log phase, lower OD values, characteristic stationary phase

~10% of growth

no growth

~15% of growth

late log phase, lower OD values, characteristic stationary phase

Comment

Data are consistent

Data are consistent

Data are consistent

fen1Δ/Δ

~40% of growth

similar growth to control

~40% of growth

late log phase, lower OD values, characteristic stationary phase

~20% of growth

similar growth to control

Comment

Data are consistent

Data are consistent

Data are consistent

fen1Δ/Δ

~50% of growth

similar growth to control

~50% of growth

late log phase, lower OD values, characteristic stationary phase

~20% of growth

late log phase

Comment

Data are consistent

Data are consistent

Data are consistent

fen1Δ/Δ

fen12Δ/Δ

~30% of growth

similar growth to control, lower OD values

~30% of growth

late log phase, lower OD values, characteristic stationary phase

~10% of growth

late log phase, lower OD values at stationary phase

Comment

Data are consistent

Data are consistent

Data are consistent

* where:

Data are consistent

Data are comparable

Data significantly differ

According to data in the table the most of the results which significantly differ between used experiments are for combination of PSZ and SU against C. albicans mutants with alterations in ergosterol level. In presence of SU with PSZ the ergosterol mutants are resistant to that combination of compounds (Figure 3), while we observed no growth during determination of growth curves. It is worth to mention that during the MIC experiment for PSZ, SU and CA C. albicans strains were incubated stationary (chapter 4.3. in materials and method section) while during analysis of growth phase C. albicans strains were cultured with shaking (chapter 4.4. in materials and method section). Therefore, we assume that detection of no growth during analysis of growth phases is caused by different culture conditions. SU prevents adhesion of C. albicans cells (Figure 9) so the agitation could additionally impact on C. albicans adhesion resulting in no significant growth in the experiment.

On the other hand, C. albicans strains fen1Δ/Δ and fen12Δ/Δ treated with PSZ alone have a similar growth curve as in control condition and this could be due to shaking which normally increases the growth rate of C. albicans when no anti-adherence agent is not present. Similar situation is when PSZ and CA mixture is used against C. albicans SC5314 and fen1Δ/Δ (growth curve is similar to control while in MIC tests we detected 15 and 20% of growth comparing to control condition, respectively) or against KS058, 10C1B1I1 and fen12Δ/Δ (significantly delayed growth rate (late logarithmic growth phase) while in MIC tests the growth of these strains is diminished to around 30 or 20%, respectively).

Therefore, we think that different cultures condition can impact on the course of growth curves. The aim of our study was to perform the broad spectrum of different tests, to properly determine the potential effect of tested combination of compound on pathogenic C. albicans strains. Thus, despite some discrepancy between MIC tests performed stationary and analysis of growth rate performed with shaking, we hope that the results show the adverse effect of PSZ with SU or CA on antifungal potential of this combination.

  1. I also have reservations about Table 1 the permeabilization (%) of the plasma membrane (PM) of C. albicans strains. At least 150 cells were analyzed in 3 independent biological repetitions for every strain in each condition. Why is the error percentage so high? It is often equal to 50% or more.

Thank you for this question. Here we presented the data (% of permeabilization) analyzed from 3 independent repetition. Therefore, we calculated the percent of permeabilization for the 3 experiments and we showed the mean and error for each condition. Here we present the example of data obtained for C. albicans SC5314 in control conditions:

Repetition

Number of cells

Number of cells with permeabilized PM

Permeabilization %

1

169

0

0.000

2

187

1

0.535

3

189

2

1.058

SUM

545

3

0.531

Ü Mean (%)

0.529

Ü Error (%)

According to you comment we decided to change the method of statistical analysis. We treated all 3 repetitions as a population (545 cells in this example) and we calculated the standard error (SE) from the population according to formula:

where:    p - % of cells with permeabilized PM in population,

               q - % of cells without permeabilized PM in population,

               n – number of cells in population.

We also implemented changes in description of Table 1 (line: 302), values in Table 1 and also in the Materials and methods section (lines: 505-510) to clarify the description of the analysis.

  1. I have a question related to the lack of data in 4.3. C. albicans susceptibility testing (determination of minimal inhibitory concentration (MIC) value) line 411. Was DMSO or another solvent used? I also have doubts whether the inoculum should not be more diluted.

The original stock of PSZ (1 mg/mL) was prepared in DMSO. The original stock of SU (1 mg/mL) and CA (45.3 mg/mL) was prepared in H2Odd and methanol, respectively. We added this information in lines: 450–453. All tested compound were prepared in serial dilutions in YPD medium using sterile 96-well plates as described in section 4.3 (lines: 450 and 457).

The OD600 of inoculum = 0.01 is routinely used in our research [1] and also in other research teams [2].

  1. The discussion should describe examples in which a compound that does not inhibit the growth of the microorganism supports the action of the fungistatic.

Thank you for this comment.  We added the proper fragment about the potentiators to discussion section (lines: 331-341).

  1. There is no explanation in the figures and tables as to what *,**, *** means.

Thank you for this remark. We implemented adequate description in lines: 110 (Figure 3), 136 (Figure 4), 154-155 (Figure 5), 170 (Figure 6), 235 (Figure 9) and 305 (Table 1).

References:

[1] Suchodolski J., Feder-Kubis J., Krasowska A. Antifungal activity of ionic liquids based on (-)-menthol: a mechanism study. Microbiol Res. 2017; 197, 56-64.

[2] Salama O. E., Gerstein A. C. Differential Response of Candida Species Morphologies and Isolates to Fluconazole and Boric Acid. Antimicrob Agents Chemother. 2022; 66(5), 0240621.
